# Association between Drinking Patterns and Incident Hypertension in Southwest China

**DOI:** 10.3390/ijerph19073801

**Published:** 2022-03-23

**Authors:** Yawen Wang, Yuntong Yao, Yun Chen, Jie Zhou, Yanli Wu, Chaowei Fu, Na Wang, Tao Liu, Kelin Xu

**Affiliations:** 1Key Laboratory of Public Health Safety, NHC Key Laboratory of Health Technology Assessment, Department of Biostatistics, School of Public Health, Fudan University, Shanghai 200032, China; 21211020135@m.fudan.edu.cn; 2Guizhou Province Centre for Disease Control and Prevention, 101 Bageyan Road, Yunyan District, Guiyang 550004, China; lovexue0716@126.com (Y.Y.); zhoujie19872014@163.com (J.Z.); wuyanli871009@163.com (Y.W.); 3Key Laboratory of Public Health Safety, NHC Key Laboratory of Health Technology Assessment, Department of Epidemiology, School of Public Health, Fudan University, Shanghai 200032, China; 18211020001@fudan.edu.cn (Y.C.); fcw@fudan.edu.cn (C.F.); na.wang@fudan.edu.cn (N.W.)

**Keywords:** alcohol consumption, cohort study, drinking frequency, hypertension, interaction effect

## Abstract

Based on a prospective cohort study of adults from southwest China with heterogeneity in their demographical characteristics and lifestyles, we aimed to explore the association between drinking patterns and incident hypertension under the interaction of these confounding factors. The Cox proportional hazard model was used to estimate the hazard ratios (HR) and 95% confidence intervals (95% CI). Subgroup analysis was performed according to sex, ethnicity, area, occupation, smoking, and exercise to compare the differences in the association between drinking patterns and the incidence of hypertension. Blood pressure was higher in participants with a high drinking frequency than those with a low drinking frequency (*p* < 0.001). We found that total drinking frequency, liquor drinking frequency, rice wine drinking frequency, and alcohol consumption were significantly associated with an increased risk of hypertension. Compared with the non-drinking group, a heavy drinking pattern was positively correlated with hypertension. Drinking can increase the risk of hypertension, especially heavy drinking patterns, with a high frequency of alcohol intake and high alcohol consumption. From the analysis results of the longitudinal data, drinking alcohol is still an important risk factor for hypertension among Chinese subjects, especially for men, the rural population, the employed, the Han nationality, smokers, and certain exercise populations.

## 1. Introduction

Hypertension seriously affects human health and quality of life, threatening the health of 1.13 billion people around the world [1]. In a nationwide survey in 2018 assessing the prevalence of hypertension in China, the investigators found a high prevalence of hypertension, with 23.2% of people aged 18 and above suffering from this health issue [2]. Alcohol consumption has been reported to be a common and modifiable risk factor for hypertension, which is supported by a number of previous findings [3].

Several studies have shown a positive association between alcohol consumption and the risk of hypertension [4,5,6]. A meta-analysis consisting of 36 trials showed that a reduction in alcohol consumption reduces blood pressure in a dose-dependent manner, with an apparent threshold effect occurring at two drinks per day [7]. Some studies have focused on the interaction effect of gender and alcohol consumption on the risk of hypertension [8,9,10,11,12]. In men, some studies have found that any alcohol consumption is associated with an increased risk of hypertension [8]; others have found that heavy drinking is associated with an increased risk of hypertension, while low and moderate drinking only have a trend toward increased risk of hypertension [9]. There appears to be no increased risk of hypertension with the consumption of 1 to 2 drinks/day, but an increased risk at higher consumption levels, in women [8]. In addition, we can further consider the association between the drinking frequency of different types of alcoholic beverages and the risk of hypertension. However, interaction effects with other risk factors such as ethnicity, area, occupation, and exercise are yet to be discussed, and our study may reveal different association characteristics between drinking patterns and incident hypertension among different people.

Therefore, based on a cohort study of adults from southwest China with heterogeneity in their demographical characteristics and lifestyles, using the Cox proportional hazards regression model and performing detailed subgroup analysis, we aimed to assess the effects of different baseline drinking indicators on the risk of hypertension, and ultimately provide a theoretical basis for the etiology of the disease.

## 2. Materials and Methods

### 2.1. Study Design and Population

The Guizhou Population Health Cohort Study (GPHCS) was a prospective cohort study based on a community in Guizhou province, China. Using the method of multi-stage proportional stratified cluster sampling, 9280 adult residents were recruited from 48 townships of 12 districts in Guizhou province from 2010 to 2012. The inclusion criteria were: (1) over 18 years of age; (2) living in the study region and having no plan to move out; (3) willingness to fill out questionnaires and undergo blood sampling; and (4) signing of written informed consent. This study was approved by the Institutional Review Board of Guizhou Province Centre for Disease Control and Prevention (No. S2017-02). All subjects provided a written informed consent form at the time of registration.

During 2016–2020, 9280 participants were followed for major chronic diseases and vital status by repeated surveys, and 1117 (12.04%) were lost to follow-up. All deaths were confirmed through the Death Registration Information System and Basic Public Health Service System. We obtained the outcome of hypertension through questionnaire follow-up and physical examination, and we excluded participants if they were lost to follow-up. We further excluded 2132 individuals with a history of hypertension at baseline and 12 individuals without blood pressure information at follow-up. Finally, the remaining 5625 subjects were eligible for the analysis.

### 2.2. Drinking Indicators

Six drinking indicators were included in this study: total drinking frequency, liquor (≥42%) drinking frequency, liquor (<42%) drinking frequency, beer drinking frequency, rice wine drinking frequency, and alcohol consumption. Here, liquor (42%) refers to the degree of alcohol, which is 42 L of ethanol per 100 L of liquor. We classified the participants into four groups according to baseline drinking frequency as follows: rarely or never drink, low-frequency drinking (monthly: <1 or 1–3 days per month), moderate-frequency drinking (weekly: 1–2 or 3–4 or 5–6 days per week) and high-frequency drinking (daily). We calculated the average of pure alcohol in g/d using the usual daily amount and frequency of alcohol consumption reported in the questionnaire in the past 12 months. The statistical form of alcohol consumption was divided into two types: volume and mass, and the frequency of alcohol consumption was divided into three forms: days per week, days per month, and days per year. The subjects selected one form to fill in according to their own specific circumstances. The degree of alcohol indicates the volume percentage of ethanol, while the intake of pure alcohol is expressed by mass. With knowledge of the degree of alcohol and drinking volume, the intake of pure alcohol can be directly derived from:*m = 0.00789·x·V*,
where *V* is the amount of alcohol consumed in milliliters, *x* is degree of alcohol, and *m* is the intake of pure alcohol (g). We can calculate the mass percentage of ethanol by combining the density of pure alcohol (789 kg/m^3^) and the density of water (1000 kg/m^3^) when the degree of alcohol and drinking quality are known, and then the intake of pure alcohol is derived from:*m = V·7.89·x/(1000 − 2.11·x)*.

The average intake of pure alcohol in grams per day (g/d) was finally obtained by combining the frequency of alcohol consumption over the past 12 months:*m′ = (52.14·n_1_ + 12·n_2_ + n_3_)·m/365*
where *m′* is the average pure alcohol intake over the past 12 months; *n_1_*, *n_2_*, *n_3_* are drinking frequencies, indicating days per week, days per month, and days per year, respectively; and *m* is the intake of pure alcohol (g). The participants were categorized into four groups according to baseline alcohol consumption: non-drinking (0 g/d), light drinking (0–12 g/d), moderate drinking (12–24 g/d), and heavy drinking (>24 g/d) [11]. Alcohol consumption, as a continuous variable, was standardized before being incorporated into the model.

### 2.3. Hypertension

Hypertension was considered present if participants met either of the following criteria: (1) self-reported doctor diagnosis of hypertension or use of anti-hypertensive medications; or (2) systolic blood pressure (SBP) of ≥140 mmHg and/or diastolic blood pressure (DBP) of ≥90 mmHg [13]. Blood pressure was evaluated in the following ways: (1) if the difference between the three measurements did not exceed 10 mmHg, the average value of three measurements should be taken as the final reading; (2) if the difference of the three measurements was large, the average value of two similar measurements would be taken as the final reading; (3) if only one measurement was obtained, it was taken as the final reading. The blood pressure was measured to within 0.1 mmHg with the same type of electronic sphygmomanometer.

### 2.4. Covariates

Participant demographical characteristics (sex, age, area, ethnicity, marital status, and occupation), lifestyle (smoking status and exercise), and history of diabetes were collected by trained investigators through face-to-face interviews using structured questionnaires. Current smoking was defined as: (1) smoking every day presently; or (2) smoking now but not every day. Participants were defined as non-current smokers as long as they did not smoke now. Participants were defined as exercise populations if they met any of the following: (1) high intensity activity with activity lasting longer than 10 min in work, farm work, and housework activities; (2) moderate intensity activity with activity lasting longer than 1 min in work, farm work, and housework activities; (3) walking or cycling for at least 10 min while outside; (4) high intensity activities that lasted at least 10 min and caused significant increases in respiration and heartbeat, such as long-distance running, swimming, or playing soccer; (5) moderate intensity exercise and leisure activities that lasted at least 10 min and caused a mild increase in breathing and heartbeat, such as brisk walking or Tai Chi. If none of the above five criteria were met, the participants were defined as no exercise. Type 2 diabetes mellitus (T2DM) was defined if participants met any of the following criteria: (1) self-reported doctor diagnosis of diabetes or use of anti-diabetic medications; (2) fasting plasma glucose (FPG) of ≥7.0 mmol/L; (3) oral glucose tolerance test (OGTT) result of ≥11.1 mmol/L; or (4) hemoglobin A1c (HbA1c) of ≥6.5% [14]. Anthropometric measurements, including height, weight, and blood pressure, were taken by trained investigators. The standing height was measured to the nearest 0.001 m with a portable tonometer. Weight was measured to the nearest 0.1 kg using a digital weighing scale. Body mass index (BMI) was calculated as body weight in kilograms divided by height in meters, squared (kg/m^2^). Venous blood samples were collected after overnight fasting for at least 8 h, and total cholesterol (TC), triglycerides (TG), high-density lipoprotein cholesterol (HDL), and low-density lipoprotein cholesterol (LDL) were measured.

### 2.5. Statistical Analysis

The continuous variables were described by the mean and standard deviation, and the categorical variables were described by frequency and percentage of participants. The number of person-years (PYs) of follow-up was calculated from the date of enrolling the cohort to the date of diagnosis of hypertension, death, or follow-up, whichever came first. The multivariate Cox proportional hazards regression model was used to analyze the hazard ratio (HR) and its 95% confidence interval (95% CI) between drinking indicators and the incidence of hypertension. We developed three separate models for the association between each drinking indicator and the risk of hypertension: (1) Model 1: adjusted for age (as continuous) and sex; (2) Model 2: Model 1 plus area (urban, rural), ethnicity (ethnic minorities, the Han nationality), marital status (married, unmarried, other), occupation (farmer, unemployed, and retired, other), smoking status (current smokers, non-current smokers), exercise (yes, no), and history of diabetes (yes, no); and (3) Model 3: Model 2 plus SBP, total cholesterol, triglycerides, HDL-C value, LDL-C value, and baseline BMI value. For variables that failed the PH test, we established time-related covariates. We divided the period and guaranteed that the PH assumption of the variable held at the two time periods. Subgroup analyses were performed by variables associated with drinking indicators, with stratification factors including sex (men, women), area (urban, rural), occupation (employed, unemployed, and retired), exercise (yes, no), smoking (yes, no), and ethnicity (ethnic minorities, the Han nationality).

To assess the robustness of the results, we estimated the association between alcohol indicators and the risk of hypertension after excluding new cases of hypertension within one year of follow-up. Then, we explored the potential for unmeasured confounding between drinking indicators and the risk of hypertension by calculating E-values [15,16]. The E-value quantifies the required magnitude of an unmeasured confounder that could negate the observed association between drinking and the risk of hypertension.

All statistical tests were two-sided and *p* < 0.05 was considered statistically significant. All analyses were performed in R software (version 4.1.0).

## 3. Results

### 3.1. Baseline Characteristics

At baseline, the age of subjects was 42.03 ± 14.17 years old. The number of women was 3062 (54.4%), and the number of Han was 3236 (57.5%). Most of the subjects were married, employed as farmers, and lived in rural areas. A small proportion had a history of diabetes and were current smokers. The group of subjects who consumed alcohol more frequently (weekly or daily) included a significantly higher proportion of current smokers. The SBP, triglycerides, total cholesterol, and HDL cholesterol all significantly correlated with the frequency of alcohol consumption in the participants (*p* < 0.001). See Table 1 for other details.

### 3.2. Associations between Drinking and Incident Hypertension

We found that total drinking frequency, liquor drinking frequency, and rice wine drinking frequency were associated with hypertension in the study population. As shown in Table 2, after adjustment for potential covariates, drinking alcohol daily was associated with an increased risk of hypertension compared with those who did not drink or rarely drank (HR: 1.29, 95% CI: 1.03–1.63). Compared with subjects whose liquor drinking frequency was rarely or never, those who drank liquor daily had a statistically increased risk of hypertension regardless of degree ≥42% or <42%, with fully adjusted HRs of 1.73 (95% CI: 1.25–2.42) and 2.24 (95% CI: 1.49–3.35), respectively. For individuals whose follow-up time was <6.6 years, drinking rice wine daily had a 58% increased risk of hypertension compared with those who drank little or no rice wine. For individuals whose follow-up time was ≥6.6 years, the risk of hypertension increased by 79% when rice wine drinking frequency was monthly, compared with subjects who did not drink rice wine or rarely drank it. For per SD increase in the amount of alcohol consumed, the risk of hypertension increased by 12% (HR = 1.12, 95% CI: 1.07–1.17) after adjusting for covariates. Compared with non-drinkers, those who engaged in heavy drinking had a statistically increased risk of hypertension, with a fully adjusted HR of 1.40 (95% CI: 1.12–1.75).

### 3.3. Subgroup Analysis

Stratification factors included sex (men, women), area (urban, rural), occupation (employed, unemployed and retired), exercise (yes, no), smoking (yes, no), and ethnicity (ethnic minority, Han). Differences in the association between drinking and the incidence of hypertension were compared. Subgroup analysis was performed for certain variables, adjusting for other confounders.

Different degrees of covariate adjustment were performed in the Cox proportional hazards regression models, in which Model 1 only adjusted for sex and age, Model 2 added area, ethnicity, marriage, occupation, smoking status, exercise, and history of diabetes, and Model 3 added SBP, total cholesterol, triglycerides, HDL-C value, LDL-C value, and baseline BMI value. In addition, some lines in the results were removed because the sample size was too small or there were no cases. Only the figures of Model 3 under the classification are presented below; the figures of Model 1 and Model 2 under the classification are provided in the Appendix A.

#### 3.3.1. Stratified by Gender

As shown in Figure 1, the left panel of the hazard ratio and 95% confidence interval represented the results in men, while the right panel represented the results in women. After stratification by gender, drinking alcohol daily, drinking liquor (≥42%) daily, drinking liquor (<42%) weekly or daily, and amount of alcohol consumed increased the risk of hypertension in men, whereas these correlations were not statistically significant in women. In men, 6.6 years was used as the limit to divide the time into two time periods: early (time < 6.6 years) and late (time ≥ 6.6 years). Drinking rice wine daily in the early stage increased the risk of hypertension in men by 98% (HR = 1.98, 95% CI: 1.19–3.30), and drinking rice wine monthly in the late stage increased the risk of hypertension in men by 175% (HR = 2.75, 95% CI: 1.65–4.58). In men, for the per SD increase in alcohol consumption, the risk of hypertension increased by 18% (HR = 1.18, 95% CI: 1.11–1.26). Compared with non-drinkers, men engaged in heavy drinking had a statistically increased risk of hypertension, with a fully adjusted HR of 1.57 (95% CI: 1.23–2.00). However, in women, we did not find significant associations between drinking and hypertension.

#### 3.3.2. Stratified by Area

As shown in Figure 2, the left panel of the hazard ratio and 95% confidence interval represents the results in urban areas, while the right panel represents the results in rural areas. After stratification by area, drinking alcohol daily increased the risk of hypertension in rural populations, whereas these correlations were not statistically significant in urban populations. In the urban population, when the time axis was divided into two stages at 6.6 years, drinking liquor (≥42%) weekly in the late-stage increased the risk of hypertension. In the rural population, drinking liquor (≥42%) daily increased the risk of hypertension. In addition, in rural areas, 6.6 years was used as the limit to divide the time into two time periods: early (time < 6.6 years) and late (time ≥ 6.6 years), and drinking liquor (<42%) daily in the early stage increased the hypertension risk. Alcohol consumption was associated with an increased risk of hypertension in the rural population (HR = 1.16, 95% CI: 1.09–1.23), and alcohol consumption also increased the risk of developing hypertension in urban populations (HR = 1.09, 95% CI: 1.01–1.18). In rural areas, heavy drinking was associated with an increased risk of hypertension compared with non-drinkers (HR: 1.47, 95% CI: 1.12–1.93). However, the correlation was not statistically significant in urban areas.

#### 3.3.3. Stratified by Ethnicity

The hazard ratio and 95% confidence interval in ethnic minorities are shown on the left panel in Figure 3, while the results for the Han nationality are shown on the right panel. After stratifying the population based on ethnic minority and Han nationality, we found that drinking liquor daily significantly increased the risk of hypertension in the Han nationality group, whereas these correlations were not statistically significant in the ethnic minority group. Alcohol consumption increased the risk of developing hypertension in the Han nationality group (HR = 1.12, 95% CI: 1.06–1.18), and alcohol consumption was associated with an increased risk of hypertension in the ethnic minority group (HR = 1.10, 95% CI: 1.02–1.19). Compared with non-drinkers, those engaged in heavy drinking had a statistically increased risk of hypertension in the Han nationality group, with a fully adjusted HR of 1.40 (95% CI: 1.05–1.88). However, we did not find a significant association between heavy drinking and hypertension in the ethnic minority group.

#### 3.3.4. Stratified by Occupation

We found that drinking alcohol daily, drinking liquor (≥42%) daily, and amount of alcohol consumed increased the risk of hypertension in the employed, whereas these correlations were not statistically significant in the unemployed and retired. In the employed, drinking liquor (<42%) daily had a statistically increased risk of hypertension, with a fully adjusted HR of 1.84 (95% CI: 1.31–2.60). When 6.6 years was used as the boundary to divide the time axis, drinking rice wine monthly in the late-stage increased the risk of hypertension in the employed. Additionally, we found that among the employed, alcohol consumption was associated with an increased risk of hypertension (HR: 1.14, 95% CI: 1.08–1.19), and compared with non-drinkers, those engaged in heavy drinking had a statistically increased risk of hypertension among the employed, with a fully adjusted HR of 1.43 (95% CI: 1.13–1.81). However, statistical significance was not found for unemployed and retired people. See Table 3 and Table 4 for other details.

#### 3.3.5. Stratified by Exercise

As shown in Figure 4, the hazard ratio and 95% confidence interval in the exercise populations are shown on the left panel, while the results for the no-exercise populations are shown in the right panel. After stratification by exercise, drinking alcohol daily increased the risk of hypertension in the no-exercise populations, while these associations were not statistically significant in the exercise populations. Compared with subjects whose liquor drinking frequency was rarely or never, the risk of hypertension increased when the frequency of liquor consumption was daily, regardless of degree ≥42% or <42% (in the no-exercise populations: liquor (≥42%): HR = 3.49, 95% CI: 1.46–8.38; liquor (<42%): HR = 6.72, 95% CI: 1.91–23.6; in the exercise populations: liquor (≥42%): HR = 1.57, 95% CI: 1.10–2.25; liquor (<42%): HR = 2.13, 95% CI: 1.38–3.28). In the exercise populations, only in individuals whose following time ≥ 6.6 years was the risk of hypertension increased by 72% when rice wine drinking frequency was monthly. Alcohol consumption was associated with an increased risk of hypertension in the exercise populations (HR = 1.11, 95% CI: 1.06–1.17), and alcohol consumption also increased the risk of developing hypertension in the no-exercise populations (HR = 1.19, 95% CI: 1.05–1.34). Compared with non-drinkers, among the exercise populations, heavy drinking was associated with an increased risk of hypertension (HR: 1.35, 95% CI: 1.07–1.71). However, the correlation was not statistically significant in the no-exercise populations.

#### 3.3.6. Stratified by Smoking

The hazard ratio and 95% confidence interval in the smokers are shown in the left panel in Figure 5, while the results for the non-smokers are shown in the right panel. After stratification by smoking, we found that drinking alcohol daily, drinking liquor (<42%) daily, and amount of alcohol consumed increased the risk of hypertension in smokers, while these associations were not statistically significant in non-smokers. In the smokers, when the time axis was divided into two stages at 6.4 years, drinking liquor (≥42%) weekly in the late stage increased the risk of hypertension. In addition, in smokers, 6.6 years was used as the limit to divide the time into two time periods: early (time < 6.6 years) and late (time ≥ 6.6 years), and drinking rice wine monthly in the late stage increased the hypertension risk. In the smokers, heavy drinking was associated with an increased risk of hypertension compared with non-drinkers (HR: 1.53, 95% CI: 1.15–2.02). However, the correlation was not statistically significant in non-smokers.

### 3.4. Sensitivity Analysis

In the sensitivity analysis conducted to assess the robustness of our results, the effect estimates of drinking on hypertension did not change substantially after excluding new cases of hypertension within one year of follow-up. See Table 5 for other details. 

We calculated E-values to assess the sensitivity to unmeasured confounding factors [17]. E-values well above the HR estimates for the measured covariates would suggest that there is an unmeasured or unknown confounder that has a substantially greater effect on the risk of hypertension than well-established covariates, which is unlikely. See Table 6 and Appendix A for other details.

## 4. Discussion

In this study, we analyzed the association between drinking patterns and the risk of hypertension based on a prospective cohort study in southwest China. We found that drinking was associated with the risk of hypertension in adults, especially heavy drinking patterns. Considering the frequency and amount of alcohol consumed, this study provided evidence that heavy drinking patterns increase the risk of developing hypertension. Furthermore, we found that subjects living in rural areas, those who were employed, exercise populations, smokers, and Han Chinese had a higher risk of hypertension. The observed associations were stronger in men than in women.

The association between drinking behavior and hypertension has been demonstrated by numerous studies [3,4,5,6,7,9,10,12,18,19,20,21,22,23]. The present study reports that heavy drinking patterns increased the risk of hypertension. Heavy drinking patterns, indicating higher levels of both alcohol consumption and frequency, have been proposed to increase hypertension risk [4,9,18,21,22]. Alternatively, previous findings demonstrated that drinking is a risk factor for hypertension [3,6,10,12,19,20,23] regardless of a heavy or lighter drinking pattern. Another review reported that moderate drinking is associated with a decreased incidence of hypertension [5]. In this study, only heavy and high-frequency drinking were significantly associated with the risk of developing hypertension.

Several studies have provided evidence that the association between drinking and the risk of developing hypertension differs among men and women; their results were not consistent [4,8,9,11,12,18,19,20,24,25]. In the current study, we found that the risk of hypertension was positively associated with heavy drinking in men, which was consistent with previous studies [4,8,9,11,18,19,20,24,25]. However, previous findings remain controversial. Of debate is whether light to moderate drinking patterns are a risk factor for hypertension in men and women. Studies have shown that there is a trend toward an increased risk of hypertension with light to moderate drinking among men [9]. In men, the risk of hypertension has been suggested to increase at a light and moderate level of alcohol consumption [4,8,11,19,24,25], whereas we found that light and moderate drinking were not significantly associated with the risk of hypertension in men. Among women, compared with many previous studies suggesting that alcohol intake is associated with the risk of hypertension [8,9,11,25,26], the present study provided no evidence to link drinking patterns with hypertension risk. Furthermore, this study provided evidence that the high drinking frequency of alcoholic beverages such as liquor is associated with a higher risk of hypertension in men, whereas these correlations were not statistically significant in women. In Chinese culture, women are not encouraged to drink alcohol [26], and women reported they drank much less than men in our study, which may partially explain why no association between drinking patterns and hypertension risk was found in women in our study.

The present study found that there was a positive association between alcohol consumption and an increased risk of hypertension among urban and rural residents, which was consistent with previous findings [27]. We also found that heavy drinking was associated with the risk of hypertension only in rural areas. This study provided evidence that the high drinking frequency of alcoholic beverages such as liquor (≥42%) was associated with a higher risk of hypertension in rural areas, whereas these correlations were not statistically significant among urban residents. In general, medical health provision in rural areas is not as good as in urban areas, and there may be problems with insufficient screening, such that disease may not be detected early and interventions are therefore not applied. The benefits of screening for hypertension have been confirmed [28,29]. In addition, the US Preventive Services Task Force (USPSTF) recommended screening for hypertension in adults aged 18 years or older [30]. Hypertension is preventable if diagnosed and treated early, which overall reduces morbidity and mortality. Therefore, as a preventable and controllable disease, timely detection of potential hypertensive patients and early intervention can be beneficial in reducing the burden of disease.

We also found that there was a positive association between alcohol consumption and an increased risk of hypertension among the Han and minority ethnic groups. This study provided evidence that, among Chinese Han people, a high drinking frequency is associated with an increased risk of hypertension only for liquor regardless of degree ≥ 42% or <42%, whereas these correlations were not statistically significant in the ethnic minorities. The finding that the associations between heavy drinking and the risk of hypertension was stronger in the Han nationality than in ethnic minorities was consistent with several previous reports [31,32,33]. The biological mechanisms that connect drinking to hypertension are complex. ALDH2 rs671 polymorphism has been proven to be closely related to the incidence of hypertension in the Asian population [31,32]. ALDH has an ALDH2 rs671 genotype mutation that results in decreased activity of ALDH, preventing the rapid conversion of toxic acetaldehyde to non-toxic acetic acid. It has been reported that ALDH mutations are less frequent and alcohol metabolism is stronger in ethnic minorities than in Han Chinese [33]. Therefore, the incidence and mortality of hypertension in the Han nationality are higher than in ethnic minorities.

We found an association between heavy drinking and an increase in hypertension risk for those living an unhealthy lifestyle with high stress, which was consistent with several previous studies [34,35,36]. Furthermore, this study provided evidence that the high drinking frequency of alcoholic beverages such as liquor is associated with a higher risk of hypertension in the employed, whereas these correlations were not statistically significant in the unemployed and retired. Work stressors are common sources of stress in adulthood [35]. The observed effect modification by occupation may be due to stress. Therefore, subjects with high stress had a higher risk of hypertension. Conversely, compared with previous findings that exercise may reduce blood pressure [37,38,39,40], this study suggested that subjects in exercise populations with heavy drinking had an increased risk of developing hypertension. This may be because we did not strictly match when we did the subgroup analysis, which means that we may still have obtained multifactorial interactions. In addition, this study provided evidence that, when considering the type of alcoholic beverages, high drinking frequency was only associated with an increased risk of hypertension for liquor among both the exercise and no-exercise populations. Furthermore, exercise exerts its effect through the sympathetic nervous system and the hypothalamic–pituitary–adrenal axis, directly affecting blood pressure and reducing it [39]. However, the effect of exercise on blood pressure may be limited, and affected by the type and intensity of exercise [37,38,39]. Therefore, further studies with a longer follow-up period and including more subjects are needed to explore these questions.

Smoking and drinking behaviors often occur concurrently [41,42]. Epidemiological studies have shown that smoking and alcohol intake are positively correlated [43,44]. In addition, in animal studies, nicotine increased ethanol intake and induced a re-emergence of ethanol-seeking behavior [44]. Therefore, smokers tend to consume more alcohol than non-smokers, which leads to a higher risk of hypertension than in non-smokers. The finding that alcohol consumption, especially heavy drinking, was more strongly associated with hypertension risk in smokers than in non-smokers were consistent with several previous reports [42,45,46]. However, this association remains controversial. Studies have shown that compared with non-drinkers and non-smokers, 154 g ethanol/week drinkers and non-smokers had a statistically increased risk of hypertension [47]. Comparatively, in the current study, we did not find that drinking increased the risk of hypertension among non-smokers.

In this study, we used a well-characterized population-based cohort with long-term follow-up in southwest China. The relatively low loss to follow-up of this cohort limited the potential bias for risk estimates. The kinds of alcoholic beverages were relatively abundant in this study, and the association results of drinking patterns and incident hypertension stratified by sex, ethnicity, area, occupation, smoking, and no exercise were studied in detail. Finally, this study was based on a cohort study of adults in southwest China, which is a region with heterogeneity in ethnicities and lifestyles. However, this study also had notable limitations. Firstly, only baseline information of most covariates was used in all analyses, which might lead to residual confounding if those covariates are time-varying. Secondly, although the current analysis adequately adjusted for major potential confounding factors, we may have ignored other risk factors affecting the incidence of hypertension, leading to the possibility of inconsistent findings. Thirdly, because there was only one follow-up, the timing of the onset of hypertension in this cohort could be inaccurate. Fourthly, our analysis should also consider the timing of exposure to further assess the effect of the cumulative exposure dose on the risk of hypertension. When the PH hypothesis did not hold, most of our results showed that the relationship between drinking and hypertension was not significant over a short period of time, but was significant over a long period of time, and the cumulative effect needs further study. Finally, whether there are differences in alcohol consumption thresholds between different categories, such as men and women [4], remains to be seen, and further study is needed.

## 5. Conclusions

Drinking can increase the risk of hypertension, especially heavy drinking patterns, with a high frequency of alcohol intake and high alcohol consumption. From the analysis results of longitudinal data, drinking alcohol is still an important risk factor for hypertension among Chinese subjects, especially for men, the rural population, the employed, the Han nationality, smokers, and certain exercise populations.

## Figures and Tables

**Figure 1 ijerph-19-03801-f001:**
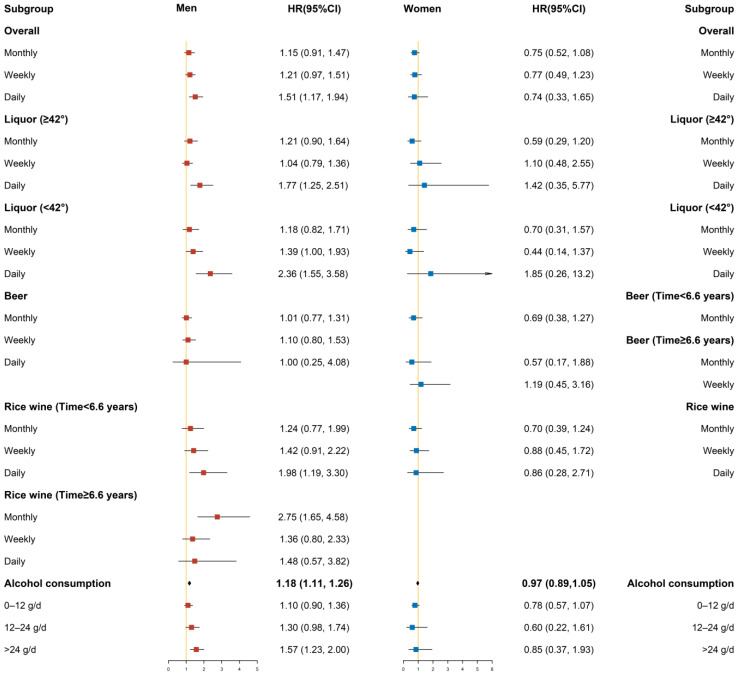
Subgroup analysis after stratification by gender (Model 3).

**Figure 2 ijerph-19-03801-f002:**
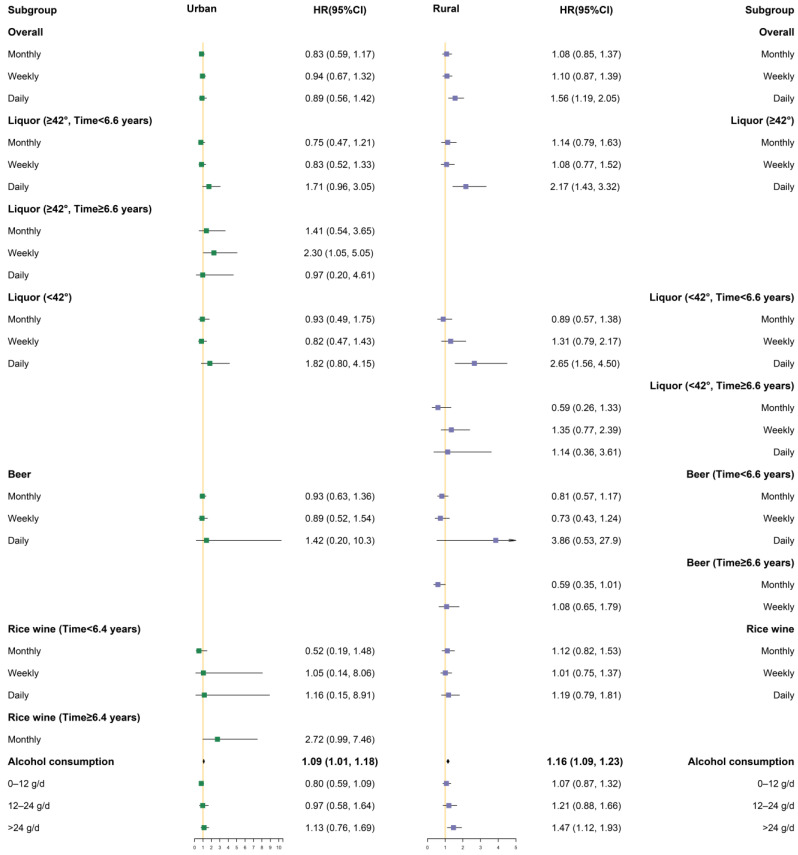
Subgroup analysis after stratification by urban and rural areas (Model 3).

**Figure 3 ijerph-19-03801-f003:**
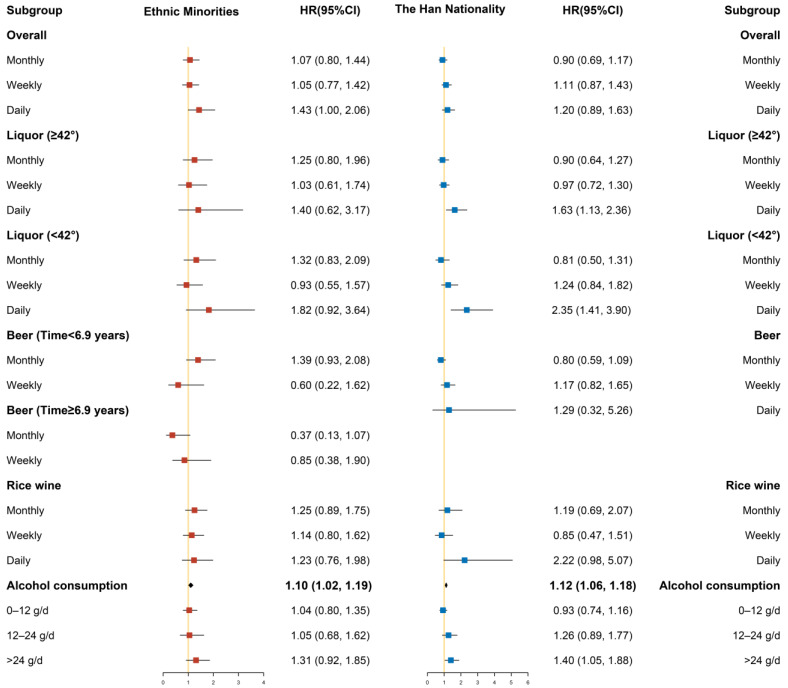
Subgroup analysis after stratification by ethnic minority and the Han nationality (Model 3).

**Figure 4 ijerph-19-03801-f004:**
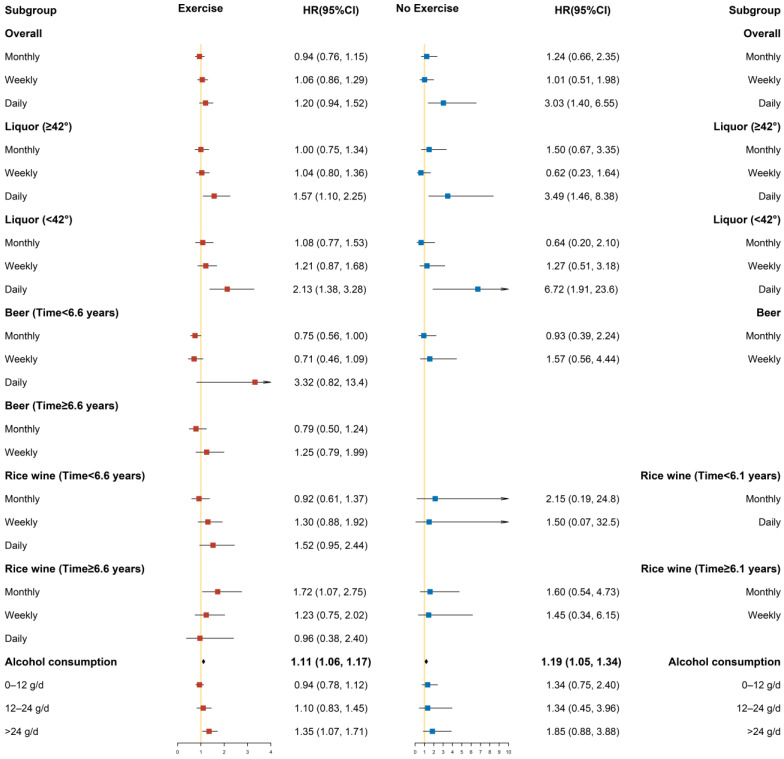
Subgroup analysis after stratification by exercise (Model 3).

**Figure 5 ijerph-19-03801-f005:**
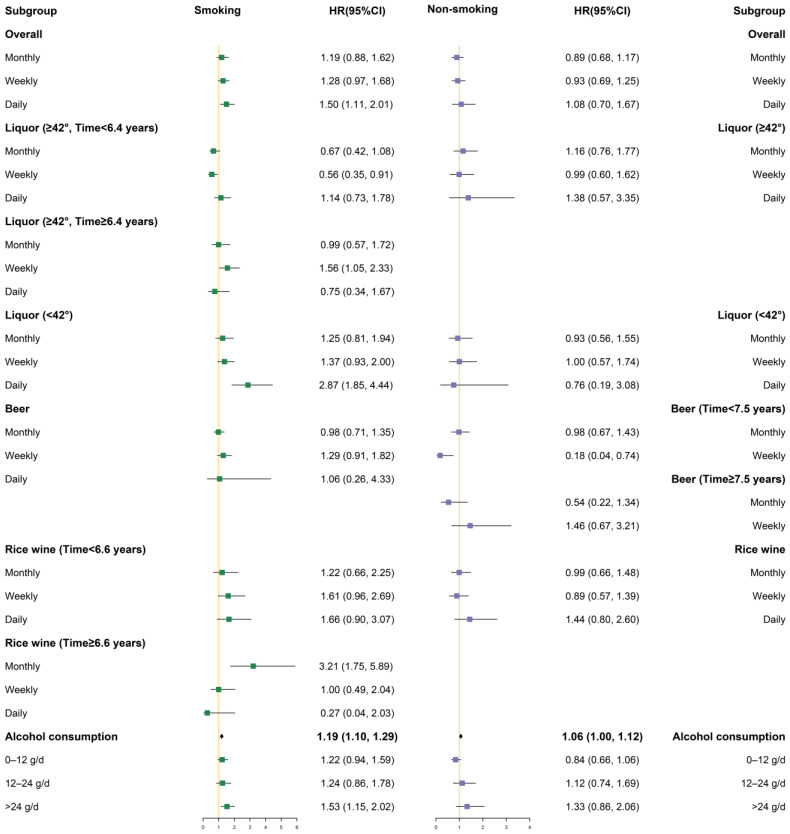
Subgroup analysis after stratification by smoking (Model 3).

**Table 1 ijerph-19-03801-t001:** Baseline characteristics according to the frequency of alcohol consumption.

	Total	Frequency of Alcohol Consumption				*p*-Value
		Rarely/Never	Monthly	Weekly	Daily	
Participants, n	5625	3888	757	641	339	
Basic indicators						
Age at baseline, years	42.03 ± 14.17	41.99 ± 14.43	39.11 ± 13.30	42.11 ± 12.93	48.95 ± 13.02	<0.001
Men, %	2563 (45.6)	1245 (32.0)	472 (62.4)	543 (84.7)	303 (89.4)	<0.001
Ethnic minority, %	2389 (42.5)	1575 (40.5)	344 (45.4)	311 (48.5)	159 (46.9)	<0.001
Rural, %	3764 (66.9)	2585 (66.5)	486 (64.2)	444 (69.3)	249 (73.5)	0.012
Marriage, %						<0.001
Married	4532 (80.6)	3158 (81.2)	564 (74.5)	510 (79.6)	300 (88.5)	
Unmarried	609 (10.8)	399 (10.3)	116 (15.3)	83 (12.9)	11 (3.2)	
Others	484 (8.6)	331 (8.5)	77 (10.2)	48 (7.5)	28 (8.3)	
Occupation, %						<0.001
Farmer	3205 (57.0)	2238 (57.6)	362 (47.8)	375 (58.5)	230 (67.8)	
Others	1594 (28.3)	1037 (26.7)	281 (37.1)	197 (30.7)	79 (23.3)	
Unemployed or retired	826 (14.7)	613 (15.8)	114 (15.1)	69 (10.8)	30 (8.8)	
Current smoker, %	1521 (27.0)	525 (13.5)	334 (44.1)	401 (62.6)	261 (77.0)	<0.001
Alcohol use, %	1737 (30.9)	0 (0.0)	757 (100.0)	641 (100.0)	339 (100.0)	<0.001
Exercise, %	4861 (86.4)	3275 (84.2)	695 (91.8)	575 (89.7)	316 (93.2)	<0.001
History of diabetes, % *	355 (6.3)	222 (5.7)	51 (6.7)	54 (8.4)	28 (8.3)	0.115
Biochemical indicators						
BMI, kg/m^2^	22.51 ± 3.16	22.47 ± 3.13	22.52 ± 2.98	22.83 ± 3.50	22.43 ± 3.12	0.058
SBP, mmHg	116.20 ± 11.94	115.58 ± 12.08	116.61 ± 11.51	117.70 ± 11.53	119.55 ± 11.28	<0.001
Triglycerides, mg/dL *	1.65 ± 1.49	1.57 ± 1.20	1.59 ± 1.24	1.87 ± 1.70	2.20 ± 3.35	<0.001
Total cholesterol, mg/dL *	4.73 ± 1.30	4.75 ± 1.30	4.47 ± 1.28	4.76 ± 1.21	4.99 ± 1.43	<0.001
HDL cholesterol, mg/dL *	1.45 ± 0.57	1.46 ± 0.57	1.37 ± 0.51	1.47 ± 0.59	1.59 ± 0.57	<0.001
LDL cholesterol, mg/dL *	2.62 ± 1.17	2.63 ± 1.20	2.49 ± 1.13	2.63 ± 1.09	2.68 ± 1.05	0.012

Note: * missing value. Abbreviations: BMI, body mass index; SBP, systolic blood pressure; HDL cholesterol, high-density lipoprotein cholesterol; LDL cholesterol, low-density lipoprotein cholesterol.

**Table 2 ijerph-19-03801-t002:** Hazard ratios (95% confidence intervals) of hypertension associated with drinking.

	Case, n	HR (95% CI)		
		Model 1	Model 2	Model 3
Frequency of alcohol consumption				
Rarely/Never	3888	1.00	1.00	1.00
Monthly	757	1.00 (0.83, 1.21)	0.98 (0.81, 1.19)	0.97 (0.80, 1.18)
Weekly	641	1.12 (0.93, 1.35)	1.10 (0.91, 1.33)	1.06 (0.88, 1.29)
Daily	339	1.35 (1.09, 1.68) **	1.31 (1.04, 1.65) *	1.29 (1.03, 1.63) *
Liquor (≥42%)				
Rarely/Never	4895	1.00	1.00	1.00
Monthly	314	1.09 (0.83, 1.41)	1.06 (0.81, 1.38)	1.05 (0.80, 1.37)
Weekly	314	1.14 (0.90, 1.46)	1.10 (0.86, 1.41)	1.02 (0.79, 1.31)
Daily	102	1.68 (1.22, 2.32) **	1.58 (1.13, 2.20) **	1.73 (1.25, 2.42) **
Liquor (<42%)				
Rarely/Never	5230	1.00	1.00	1.00
Monthly	187	1.13 (0.82, 1.56)	1.12 (0.81, 1.55)	1.05 (0.76, 1.47)
Weekly	155	1.29 (0.95, 1.76)	1.24 (0.91, 1.69)	1.20 (0.88, 1.63)
Daily	53	2.43 (1.63, 3.64) ***	2.35 (1.57, 3.52) ***	2.24 (1.49, 3.35) ***
Beer				
Rarely/Never	4945	1.00	1.00	1.00
Monthly	456	0.96 (0.76, 1.20)	0.93 (0.74, 1.18)	0.94 (0.74, 1.19)
Weekly	215	1.07 (0.80, 1.43)	1.07 (0.79, 1.44)	1.05 (0.78, 1.41)
Daily	9	1.00 (0.25, 4.02)	1.09 (0.27, 4.39)	1.00 (0.25, 4.03)
Rice wine (Time < 6.6 years)				
Rarely/Never	2575	1.00	1.00	1.00
Monthly	125	1.05 (0.73, 1.51)	0.95 (0.65, 1.38)	0.97 (0.66, 1.42)
Weekly	93	1.40 (0.97, 2.01)	1.15 (0.79, 1.67)	1.13 (0.77, 1.67)
Daily	47	1.82 (1.17, 2.82) **	1.58 (1.01, 2.49) *	1.58 (1.00, 2.50) *
Rice wine (Time ≥ 6.6 years)				
Rarely/Never	2433	1.00	1.00	1.00
Monthly	130	2.23 (1.46, 3.41) ***	1.82 (1.17, 2.82) **	1.79 (1.15, 2.80) *
Weekly	164	1.62 (1.03, 2.54) *	1.37 (0.86, 2.17)	1.28 (0.81, 2.05)
Daily	58	1.57 (0.64, 3.85)	1.24 (0.50, 3.06)	1.06 (0.43, 2.65)
Alcohol consumption				
	5616	1.12 (1.07, 1.17) ***	1.11 (1.07, 1.16) ***	1.12(1.07, 1.17) ***
0 g/d	3931	1.00	1.00	1.00
0–12 g/d	1034	1.01 (0.86, 1.19)	0.98 (0.83, 1.16)	0.97 (0.82, 1.15)
12–24 g/d	262	1.19 (0.92, 1.55)	1.17 (0.90, 1.52)	1.13 (0.86, 1.48)
>24 g/d	389	1.45 (1.18, 1.79) ***	1.43 (1.15, 1.78) **	1.40 (1.12, 1.75) **

Note: Model 1: adjusted for age (continuous variable), sex; Model 2: Model 1 plus area, ethnicity, marriage, occupation, smoking status, exercise, and history of diabetes; Model 3: Model 2 plus SBP, total cholesterol, triglycerides, HDL-C value, LDL-C value, baseline BMI value; ***: *p* < 0.001, **: *p* < 0.01, *: *p* < 0.05. Abbreviations: HR, hazard ratio; 95% CI, 95% confidence interval.

**Table 3 ijerph-19-03801-t003:** Hazard ratios (95% confidence intervals) of hypertension associated with frequency of alcohol consumption among the employed.

	Case, n	HR (95% CI)		
		Model 1	Model 2	Model 3
Frequency of alcohol consumption				
Rarely/Never	3275	1.00	1.00	1.00
Monthly	643	0.99 (0.81, 1.22)	0.99 (0.80, 1.22)	0.98 (0.80, 1.22)
Weekly	572	1.11 (0.91, 1.35)	1.11 (0.90, 1.36)	1.08 (0.88, 1.33)
Daily	309	1.38 (1.10, 1.73) **	1.39 (1.10, 1.77) **	1.38 (1.08, 1.76) **
Liquor (≥42%)				
Rarely/Never	4148	1.00	1.00	1.00
Monthly	276	1.14 (0.86, 1.50)	1.12 (0.85, 1.49)	1.12 (0.84, 1.49)
Weekly	281	1.15 (0.88, 1.49)	1.11 (0.85, 1.45)	1.01 (0.77, 1.33)
Daily	94	1.78 (1.27, 2.48) ***	1.68 (1.19, 2.36) **	1.84 (1.31, 2.60) ***
Liquor (<42%)				
Rarely/Never	4450	1.00	1.00	1.00
Monthly	166	1.05 (0.74, 1.50)	1.06 (0.75, 1.50)	1.00 (0.70, 1.43)
Weekly	136	1.27 (0.91, 1.77)	1.20 (0.86, 1.68)	1.16 (0.83, 1.62)
Daily	47	2.42 (1.59, 3.68) ***	2.45 (1.61, 3.74) ***	2.29 (1.50, 3.50) ***
Beer				
Rarely/Never	4221	1.00	1.00	1.00
Monthly	387	0.92 (0.72, 1.18)	0.89 (0.70, 1.15)	0.91 (0.71, 1.18)
Weekly	186	1.01 (0.73, 1.39)	1.01 (0.73, 1.40)	1.00 (0.72, 1.39)
Daily	5	1.78 (0.44, 7.16)	1.97 (0.49, 7.97)	1.84 (0.46, 7.47)
Rice wine (Time < 6.6 years)				
Rarely/Never	2176	1.00	1.00	1.00
Monthly	114	1.04 (0.70, 1.53)	0.92 (0.62, 1.37)	0.95 (0.63, 1.42)
Weekly	87	1.42 (0.97, 2.07)	1.18 (0.80, 1.74)	1.17 (0.79, 1.74)
Daily	44	1.76 (1.10, 2.79) *	1.56 (0.97, 2.51)	1.55 (0.95, 2.51)
Rice wine (Time ≥ 6.6 years)				
Rarely/Never	2054	1.00	1.00	1.00
Monthly	120	2.24 (1.43, 3.50) ***	1.84 (1.16, 2.93) **	1.80 (1.13, 2.87) *
Weekly	150	1.59 (0.99, 2.56)	1.37 (0.84, 2.23)	1.27 (0.78, 2.09)
Daily	54	1.56 (0.64, 3.84)	1.28 (0.51, 3.17)	1.09 (0.44, 2.73)
Alcohol consumption				
	4792	1.13 (1.08, 1.18) ***	1.13 (1.08, 1.18) ***	1.14 (1.08, 1.19) ***
0 g/d	3311	1.00	1.00	1.00
0–12 g/d	886	0.98 (0.82, 1.17)	0.96 (0.80, 1.16)	0.96 (0.80, 1.15)
12–24 g/d	241	1.22 (0.93, 1.60)	1.23 (0.93, 1.62)	1.18 (0.90, 1.57)
>24 g/d	354	1.45 (1.17, 1.81) ***	1.46 (1.16, 1.84) **	1.43 (1.13, 1.81) ***

Note: Model 1: adjusted for age (continuous variable), sex; Model 2: Model 1 plus location, nation, marriage, occupation, smoking status, exercise, and history of diabetes. Model 3: Model 2 plus SBP, total cholesterol, triglycerides, HDL-C, LDL-C value, baseline BMI value. ***: *p* < 0.001, **: *p* < 0.01, *: *p* < 0.05; Abbreviations: HR, hazard ratio; 95% CI, 95% confidence interval.

**Table 4 ijerph-19-03801-t004:** Hazard ratios (95% confidence intervals) of hypertension associated with drinking among the unemployed and retired.

	Cases, n	HR (95% CI)		
		Model 1	Model 2	Model 3
Frequency of alcohol consumption				
Rarely/Never	613	1.00	1.00	1.00
Monthly	114	1.08 (0.67, 1.74)	0.97 (0.59, 1.59)	0.98 (0.59, 1.61)
Weekly	69	1.22 (0.74, 2.04)	0.95 (0.55, 1.65)	0.92 (0.53, 1.62)
Daily	30	1.06 (0.50, 2.23)	0.86 (0.40, 1.85)	0.92 (0.42, 1.98)
Liquor (≥42%)				
Rarely/Never	747	1.00	1.00	1.00
Monthly	38	0.84 (0.37, 1.91)	0.78 (0.34, 1.81)	0.82 (0.35, 1.90)
Weekly	33	1.29 (0.68, 2.42)	1.05 (0.54, 2.04)	1.05 (0.53, 2.06)
Daily	8	0.83 (0.20, 3.41)	0.65 (0.16, 2.71)	0.70 (0.17, 2.95)
Liquor (<42% Time < 6.3 years)				
Rarely/Never	285	1.00	1.00	1.00
Monthly	10	1.88 (0.75, 4.71)	1.41 (0.51, 3.88)	1.32 (0.48, 3.66)
Weekly	6	0.58 (0.08, 4.23)	0.52 (0.07, 4.00)	0.36 (0.04, 2.89)
Daily	4	1.94 (0.46, 8.18)	1.82 (0.41, 8.20)	1.43 (0.31, 6.68)
Liquor (<42% Time ≥ 6.3 years)				
Rarely/Never	555	1.00	1.00	1.00
Monthly	11	0.62 (0.08, 4.50)	0.56 (0.07, 4.18)	0.45 (0.06, 3.47)
Weekly	13	2.77 (1.08, 7.07) *	2.30 (0.88, 6.02)	2.49 (0.90, 6.86)
Beer				
Rarely/Never	724	1.00	1.00	1.00
Monthly	69	1.22 (0.66, 2.25)	1.16 (0.62, 2.18)	1.09 (0.58, 2.05)
Weekly	29	1.55 (0.75, 3.19)	1.35 (0.63, 2.93)	1.29 (0.59, 2.80)
Rice wine				
Rarely/Never	778	1.00	1.00	1.00
Monthly	21	1.07 (0.44, 2.61)	0.86 (0.34, 2.14)	1.00 (0.40, 2.50)
Weekly	20	0.90 (0.33, 2.44)	0.56 (0.17, 1.80)	0.59 (0.18, 1.93)
Daily	7	1.40 (0.34, 5.76)	0.94 (0.22, 4.02)	0.95 (0.22, 4.08)
Alcohol consumption				
	824	1.04 (0.88, 1.22)	0.98 (0.80, 1.20)	0.98 (0.80, 1.21)
0 g/d	620	1.00	1.00	1.00
0–12 g/d	148	1.24 (0.83, 1.84)	1.10 (0.72, 1.68)	1.11 (0.72, 1.71)
12–24 g/d	21	0.94 (0.38, 2.34)	0.76 (0.30, 1.95)	0.72 (0.28, 1.89)
>24 g/d	35	1.33 (0.68, 2.60)	1.00 (0.48, 2.06)	0.99 (0.47, 2.06)

Note: Model 1: adjusted for age (continuous variable), sex; Model 2: Model 1 plus location, nation, marriage, occupation, smoking status, exercise, and history of diabetes. Model 3: Model 2 plus SBP, total cholesterol, triglycerides, HDL-C, LDL-C value, baseline BMI value. The frequency of liquor (<42%, time ≥ 6.3 years) consumption was daily: Two lines were removed because the sample size was too small. The frequency of beer consumption was daily: this line was removed because the sample size was too small. Abbreviations: HR, hazard ratio; 95% CI, 95% confidence interval.

**Table 5 ijerph-19-03801-t005:** Hazard ratios (95% confidence intervals) of hypertension associated with drinking after excluding new cases of hypertension within one year of follow-up.

	Cases, n	HR (95% CI)		
		Model 1	Model 2	Model 3
Frequency of alcohol consumption				
Rarely/Never	3862	1.00	1.00	1.00
Monthly	755	1.01 (0.83, 1.22)	0.99 (0.81, 1.20)	0.98 (0.81, 1.19)
Weekly	635	1.10 (0.91, 1.33)	1.07 (0.89, 1.30)	1.04 (0.86, 1.27)
Daily	338	1.38 (1.10, 1.71) **	1.33 (1.05, 1.67) *	1.31 (1.04, 1.65) *
Liquor (≥42%)				
Rarely/Never	4862	1.00	1.00	1.00
Monthly	314	1.12 (0.86, 1.45)	1.08 (0.83, 1.41)	1.07 (0.82, 1.40)
Weekly	313	1.16 (0.91, 1.48)	1.11 (0.87, 1.43)	1.02 (0.79, 1.32)
Daily	101	1.69 (1.22, 2.34) **	1.57 (1.12, 2.19) **	1.73 (1.23, 2.41) **
Liquor (<42%)				
Rarely/Never	5198	1.00	1.00	1.00
Monthly	186	1.13 (0.81, 1.56)	1.12 (0.81, 1.55)	1.05 (0.75, 1.47)
Weekly	153	1.26 (0.92, 1.73)	1.21 (0.88, 1.66)	1.16 (0.85, 1.60)
Daily	53	2.52 (1.68, 3.76) ***	2.42 (1.61, 3.62) ***	2.30 (1.53, 3.45) ***
Beer				
Rarely/Never	4915	1.00	1.00	1.00
Monthly	452	0.93 (0.73, 1.17)	0.90 (0.71, 1.14)	0.90 (0.71, 1.15)
Weekly	214	1.06 (0.79, 1.43)	1.06 (0.78, 1.43)	1.04 (0.77, 1.40)
Daily	9	1.01 (0.25, 4.07)	1.11 (0.27, 4.46)	1.01 (0.25, 4.09)
Rice wine (Time < 6.6 years)				
Rarely/Never	2541	1.00	1.00	1.00
Monthly	125	1.10 (0.76, 1.59)	1.00 (0.69, 1.45)	1.02 (0.70, 1.50)
Weekly	92	1.43 (0.99, 2.07)	1.18 (0.81, 1.73)	1.19 (0.80, 1.77)
Daily	47	1.93 (1.24, 3.00) **	1.68 (1.07, 2.65) *	1.71 (1.08, 2.71) *
Rice wine (Time ≥ 6.6 years)				
Rarely/Never	2433	1.00	1.00	1.00
Monthly	130	2.23 (1.46, 3.41) ***	1.82 (1.17, 2.82) **	1.79 (1.15, 2.80) *
Weekly	164	1.62 (1.03, 2.54) *	1.37 (0.86, 2.17)	1.28 (0.81, 2.05)
Daily	58	1.57 (0.64, 3.85)	1.24 (0.50, 3.06)	1.06 (0.43, 2.65)
Alcohol consumption				
	5581	1.12 (1.08, 1.17) ***	1.12 (1.07, 1.17) ***	1.13 (1.08, 1.18) ***
0 g/d	3905	1.00	1.00	1.00
0–12 g/d	1028	1.00 (0.85, 1.18)	0.97 (0.82, 1.15)	0.96 (0.81, 1.14)
12–24 g/d	261	1.20 (0.93, 1.57)	1.17 (0.90, 1.53)	1.14 (0.87, 1.49)
>24 g/d	387	1.46 (1.18, 1.81) ***	1.43 (1.14, 1.78) **	1.40 (1.12, 1.76) **

Note: Model 1: adjusted for age (continuous variable), sex; Model 2: Model 1 plus area, ethnicity, marriage, occupation, smoking status, exercise, and history of diabetes; Model 3: Model 2 plus SBP, total cholesterol, triglycerides, HDL-C, LDL-C value, baseline BMI value; ***: *p* < 0.001, **: *p* < 0.01, *: *p* < 0.05. Abbreviations: HR, hazard ratio; 95% CI, 95% confidence interval.

**Table 6 ijerph-19-03801-t006:** E-values for the effect of drinking on hypertension (and its lower limit of 95% CI) in each adjusted Cox model.

Model	E-Value for HR Estimate	E-Value for Lower Limit of 95% CI	Variable	Level	HR (95% CI)
Model 1	1.76	1.32	Frequency of alcohol consumption	Daily vs. Rarely/Never	1.35 (1.09, 1.68)
	2.22	1.56	Liquor (≥42%)	Daily vs. Rarely/Never	1.68 (1.22, 2.32)
	3.08	2.15	Liquor (<42%)	Daily vs. Rarely/Never	2.43 (1.63, 3.64)
	2.39	1.47	Rice wine (Time < 6.6 years)	Daily vs. Rarely/Never	1.82 (1.17, 2.82)
	2.87	1.92	Rice wine (Time ≥ 6.6 years)	Monthly vs. Rarely/Never	2.23 (1.46, 3.41)
	2.14	1.17		Weekly vs. Rarely/Never	1.62 (1.03, 2.54)
	1.38	1.27	Alcohol consumption	Alcohol consumption	1.12 (1.07, 1.17)
	1.72	1.36		>24 g/d vs. 0 g/d	1.45 (1.18, 1.79)
Model 2	1.70	1.20	Frequency of alcohol consumption	Daily vs. Rarely/Never	1.31 (1.04, 1.65)
	2.09	1.40	Liquor (≥42%)	Daily vs. Rarely/Never	1.58 (1.13, 2.20)
	3.00	2.07	Liquor (<42%)	Daily vs. Rarely/Never	2.35 (1.57, 3.52)
	2.09	1.09	Rice wine (Time < 6.6 years)	Daily vs. Rarely/Never	1.58 (1.01, 2.49)
	2.39	1.47	Rice wine (Time ≥ 6.6 years)	Monthly vs. Rarely/Never	1.82 (1.17, 2.82)
	1.36	1.27	Alcohol consumption	Alcohol consumption	1.11 (1.07, 1.16)
	1.88	1.44		>24 g/d vs. 0 g/d	1.43 (1.15, 1.78)
Model 3	1.67	1.17	Frequency of alcohol consumption	Daily vs. Rarely/Never	1.29 (1.03, 1.63)
	2.28	1.61	Liquor (≥42%)	Daily vs. Rarely/Never	1.73 (1.25, 2.42)
	2.88	1.96	Liquor (<42%)	Daily vs. Rarely/Never	2.24 (1.49, 3.35)
	2.09	1.00	Rice wine (Time < 6.6 years)	Daily vs. Rarely/Never	1.58 (1.00, 2.50)
	2.35	1.44	Rice wine (Time ≥ 6.6 years)	Monthly vs. Rarely/Never	1.79 (1.15, 2.80)
	1.38	1.27	Alcohol consumption	Alcohol consumption	1.12 (1.07, 1.17)
	1.84	1.38		>24 g/d vs. 0 g/d	1.40 (1.12, 1.75)

Note: Model 1: adjusted for age (continuous variable), sex; Model 2: Model 1 plus area, ethnicity, marriage, occupation, smoking status, exercise, and history of diabetes; Model 3: Model 2 plus SBP, total cholesterol, triglycerides, HDL-C, LDL-C value, baseline BMI value. Abbreviations: HR, hazard ratio; 95% CI, 95% confidence interval.

## Data Availability

Data can be made available to interested researchers upon reasonable request.

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
