# Peer review of "Association between Drinking Patterns and Incident Hypertension in Southwest China"

_ijerph, 2022, doi:10.3390/ijerph19073801_

Round 1
Reviewer 1 Report
I commend the authors for their study titled, " Association between drinking pattern and incident hypertension in Southwest China"
The association between alcohol consumption and hypertension is well established long ago and such information is abundantly available online in the public domain. The significance of reproducing similar studies in different environments has yet to be determined. 1
Moreover, in the methods section, the authors state about three BP measurements and it is not clear if blood pressure measurements were taken as per WHO guidelines: “Hypertension is diagnosed if, when it is measured on two different days, the systolic blood pressure readings on both days is ≥140 mmHg and/or the diastolic blood pressure readings on both days is ≥90 mmHg”.2
The definition of hypertension has also evolved with lower cut-off points now considered both for systolic and diastolic pressure.3 The current definition of hypertension is systolic blood pressure values of 130mmHg or more and/or diastolic blood pressure more than 80 mmHg.3
I do not see any new information that is produced by his study that would influence the clinical decision-making in the treatment of hypertension or public health interventions for hypertension risk reduction
References
- Yoo, M. G., Park, K. J., Kim, H. J., Jang, H. B., Lee, H. J., & Park, S. I. (2019). Association between alcohol intake and incident hypertension in the Korean population. Alcohol, 77, 19-25.
- World health organization. (2021). Hypertension: Key facts. Accessed at https://www.who.int/news-room/fact-sheets/detail/hypertension
- Centers for Disease Control.(2021). High Blood Pressure. Accessed at https://www.cdc.gov/bloodpressure/facts.htm
- Iqbal AM, Jamal SF. Essential Hypertension. [Updated 2021 Jul 26]. In: StatPearls [Internet]. Treasure Island (FL): StatPearls Publishing; 2022 Jan-. Available from: https://www.ncbi.nlm.nih.gov/books/NBK539859/
Author Response
Thank you for the comment. The important role of alcohol consumption played in hypertension has been widely discussed in the literature. However, its interaction effects with other risk factors are yet to be discussed. In this study, we explored the interaction effects on hypertension based on a cohort in southwest China, which is wealthy in the diversity of ethnicity and lifestyle. We found that subjects who were men, living in rural areas, being employed, being sporting populations, and being Han Chinese had a higher risk of hypertension. We revised our paper and highlighted these points in the sections of introduction and discussion. As for the definition of hypertension, this study referred to its traditional definition, which was widely used in cohort studies [1-4]. However, we agree that the definition of hypertension has evolved with lower cut-off points considered both for systolic and diastolic pressure. It’s an interesting point that we will explore in the subsequent studies and investigate the differences.
References
- Cardiovascular risk factors in China: a nationwide population-based cohort study [https://pubmed.ncbi.nlm.nih.gov/33271080/]
- Peng M, Wu S, Jiang X, Jin C, Zhang W: Long-term alcohol consumption is an independent risk factor of hypertension development in northern China: evidence from Kailuan study. J Hypertens 2013, 31(12):2342-2347.
- Sesso HD, Cook NR, Buring JE, Manson JE, Gaziano JM: Alcohol consumption and the risk of hypertension in women and men. Hypertension 2008, 51(4):1080-1087.
- Unger T, Borghi C, Charchar F, Khan NA, Poulter NR, Prabhakaran D, Ramirez A, Schlaich M, Stergiou GS, Tomaszewski M et al: 2020 International Society of Hypertension Global Hypertension Practice Guidelines. Hypertension 2020, 75(6):1334-1357.
Reviewer 2 Report
Line 77: liquor (≥42%) drinking frequency, liquor (<42%) drinking frequency: does 42% mean the content of ethanol? Please make sure. Also, how did the authors deal with the content of ethanol when calculating the average of pure alcohol? It would be good if the authors could explain more details.
The authors mentioned cox model was used to examine the association of drinking patterns and incident hypertension. How did the authors follow up on incident hypertension? Were there subsequent visits after baseline (2010-2012)? What if some participants did not show up for the following visits?
What was the justification for Model 2 and Model 3? I assumed Model 2 included variables from the questionnaire, and Model 3 had laboratory and physical variables. But, the authors obtained the information on diabetes from questionnaires or laboratory measures. I recommend justifying models, for example, adjusting for conventional cardiovascular risk factors and socioeconomic status/lifestyle factors.
Subgroup analysis: Heavy drinkers tend to smoke heavily. How about stratifying smoking status? Also, could the authors describe how to obtain the information on sport? It would not be clear to the readers.
The first paragraph under 3.1 Study participants should be under 2.1 Study design and participants. Then, the authors should describe how to follow up the outcome of interest in the Method section. Also, please change the subtitle of 3.1 Study participants to “baseline characteristics.”
Why was the rice wine categorized into two groups <6.6 years and +6.6 years?
In table 2, I am not sure about the authors’ interests (Frequency? Types of alcohol? Amount?). It is hard to digest the results. This might be related to subgroup analysis. The authors have a bunch of figures. I recommend choosing the best measure of alcohol consumption and showing the results from subgroup analysis in a single figure. Other measures that the authors want to mention can be moved to supplemental data or just mentioned in the test with data not shown.
For subgroup analysis, have the authors tested the interaction? If the interaction was not statistically significant, we could not simply say the association was stronger in men than women.
Line 332: How did the authors define heavy drinking patterns?
Author Response
Line 77: liquor (≥42%) drinking frequency, liquor (<42%) drinking frequency: does 42% mean the content of ethanol? Please make sure. Also, how did the authors deal with the content of ethanol when calculating the average of pure alcohol? It would be good if the authors could explain more details.
Thank you for the suggestions. 42% means that 100 volume units of liquor contain 42 volume units of ethanol. We added this point in our revised manuscript “We calculated the average of pure alcohol in g/d using the usual daily amount and frequency of alcohol consumption reported in the questionnaire in the past twelve months. The statistical form of alcohol consumption was divided into two types: volume and mass, and the frequency of alcohol consumption was divided into three forms: day/week, day/month, and day/year, and the subjects selected one form to fill in according to their specific circumstances. The degree of alcohol indicates the volume percentage of ethanol, while the intake of pure alcohol is expressed by mass. With knowledge of the degree of alcohol and drinking volume, the intake of pure alcohol can be directly derived from m=0.00789·x·V, where V is the amount of alcohol consumed in ml, x is the degree of alcohol, and m is the intake of pure alcohol (g). We can calculate the mass percentage of ethanol by combining the density of pure alcohol (789 kg/m ³) and the density of water (1000 kg/m ³) when the degree of alcohol and drinking quality is known, and then the intake of pure alcohol is derived from m=M·7.89·x/(1000-2.11·x). The average of pure alcohol in g/d was finally obtained by combining the frequency of alcohol consumption over the past twelve months: m’= (52.14·n1+12·n2+n3) ·m/365 where m’ is the average pure alcohol intake over the past 12 months, n1, n2, n3 are drinking frequencies, indicating day/week, day/month, day/year, respectively, and m is the intake of pure alcohol (g). ” For example, assuming that 42% liquor with the volume of 100ml contains 42ml of pure alcohol and 58ml of water, it is converted into 33.138g of pure alcohol and 58g of water, with the total mass of 91.138g, the mass percentage of pure alcohol in 42% liquor is 36% by dividing the mass of pure alcohol by the total mass.
The authors mentioned cox model was used to examine the association of drinking patterns and incident hypertension. How did the authors follow up on incident hypertension? Were there subsequent visits after baseline (2010-2012)? What if some participants did not show up for the following visits?
Thank you for the comment. All participants were followed up for major chronic diseases and vital status by a repeated investigation during 2016-2020, and 1117 (12.04%) were lost to follow-up. All deaths were confirmed through the Death Registration Information System and Basic Public Health Service System. We obtained the outcome of hypertension through questionnaire follow-up and physical examination, and excluded participants if they were lost to follow-up. We revised the statement in study design and population part to make this clear in the paper.
What was the justification for Model 2 and Model 3? I assumed Model 2 included variables from the questionnaire, and Model 3 had laboratory and physical variables. But the authors obtained the information on diabetes from questionnaires or laboratory measures. I recommend justifying models, for example, adjusting for conventional cardiovascular risk factors and socioeconomic status/lifestyle factors.
Thank you for your insightful comments on this issue. We first adjusted for sex and age in model 1, and then added other demographic characteristics, lifestyle, and disease history to model 1 to build model 2, and finally added biochemical indicators to model 2 to build model 3. The main results in our paper were based on model3. We also tried what you recommended in the analysis, and the target estimates of drinking effects on hypertension did not change.
Subgroup analysis: Heavy drinkers tend to smoke heavily. How about stratifying smoking status? Also, could the authors describe how to obtain the information on sport? It would not be clear to the readers.
Thanks for your insightful suggestions. We performed subgroup analysis on smoking and added a subsection “3.3.6 Stratified by smoking” and discussed the results in the Discussion part in the revised manuscript. We added how to obtain the information on sport in our revised manuscript as, “Participants were defined as sporting if they met one of the following: 1) high-intensity activity with activity lasting longer than 10 minutes in work, farm work and housework activities; 2) moderate-intensity activity with activity lasting longer than 1 minutes in work, farm work, and housework activities; 3) walking or cycling for at least 10 minutes while outside; 4) high-intensity activities that last at least 10 minutes and cause significant increases in respiration, heartbeat, such as long-distance running, swimming, playing soccer; 5) moderate-intensity exercise and leisure activities that last at least 10 minutes and cause a mild increase in breathing, heartbeat, such as brisk walking, playing Tai Chi.”
The first paragraph under 3.1 Study participants should be under 2.1 Study design and participants. Then, the authors should describe how to follow up the outcome of interest in the Method section. Also, please change the subtitle of 3.1 Study participants to “baseline characteristics.”
Thank you very much for your careful reading. We have realigned the paragraph order and changed the subtitle of 3.1. We have added how to follow up the outcome of hypertension in the revised manuscript as, “During 2016-2020, 9280 participants were followed for major chronic diseases and vital status by repeated surveys, and 1117 (12.04%) were lost to follow-up. All deaths were confirmed through the Death Registration Information System and Basic Public Health Service System. We obtained the outcome of hypertension through questionnaire follow-up and physical examination, and excluded participants if they were lost to follow-up.”
Why was the rice wine categorized into two groups <6.6 years and +6.6 years?
Thank you for the comment. Since the variable of rice wine failed the PH test, we established time-related covariates. We divided the period according to Schoenfeld residual plots and guaranteed that the PH assumption of the variable holds at the two time periods.
In table 2, I am not sure about the authors’ interests (Frequency? Types of alcohol? Amount?). It is hard to digest the results. This might be related to subgroup analysis. The authors have a bunch of figures. I recommend choosing the best measure of alcohol consumption and showing the results from subgroup analysis in a single figure. Other measures that the authors want to mention can be moved to supplemental data or just mentioned in the test with data not shown.
Thanks for your comment. For the present study, we aim to assess the effects of different baseline drinking indicators on the risk of hypertension based on a cohort study of adults from Southwest China. We investigate the impact of alcohol consumption and frequency of consumption of different types of alcoholic beverages on the incidence of hypertension to see how different drinking patterns influence the incidence of hypertension.
For subgroup analysis, have the authors tested the interaction? If the interaction was not statistically significant, we could not simply say the association was stronger in men than women.
Thank you for the comment. We compared confidence intervals to test the interaction. The purpose of the confidence interval is to avoid the uncertainty of the point estimate and to express a fact more accurately by interval estimation. If the confidence interval contains 1, we can consider that the studied factor and the outcome are not statistically related. If the confidence interval is greater than 1, we can consider that the studied factor is conducive to the occurrence of the outcome and is a risk factor for the disease. If the confidence interval is less than 1, we can consider that the studied factor is not conducive to the occurrence of the outcome and is a protective factor for the disease. To test whether the association was stronger in men than in women, we can simply compare the two confidence intervals in men and women. If the confidence interval is greater than 1 in men but contains 1 in women, and the minimum value of the confidence interval in men is greater than the maximum value of confidence interval in women, we can say the association was stronger in men than in women. In the present study, we found that the association between alcohol consumption and hypertension risk was stronger in men than in women.
Line 332: How did the authors define heavy drinking patterns?
Thank you for the comment. We classified the participants into four groups according to baseline drinking frequency as follows: rarely or never drinking, low-frequency drinking (monthly: <1 or 1-3 day per month), moderate-frequency drinking (weekly: 1-2 or 3-4 or 5-6 day per week) and high-frequency drinking (daily). In addition, participants were categorized into four groups according to baseline alcohol consumption: non-drinking (0 g/d), light drinking (0-12 g/d), moderate drinking (12-24 g/d), heavy drinking (>24 g/d). Considering the frequency and amount of alcohol consumed, we defined heavy drinking patterns as high-frequency drinking and heavy drinking.
Reviewer 3 Report
The authors analyze the data stratified by multiple variables. This is very interesting, but these subgroups increase the statistical power. The authors should discuss this aspect.
Author Response
Thank you for your insightful comment. To show the statistical power of the tests, we draw a scatterplot of the value of statistical power versus sample size. We calculated power based on sample size, effect size, and significance level. As for the sample size, we select the smaller number in the two comparing groups to guarantee the true power was larger than what we represent. Please see the attachment for the scatterplot. From this figure, we can see that when the sample size in each group is larger than 400, the power is larger than 0.6. In our subgroup analysis, most sample size in each group is larger than 1000, which means our subgroup analysis can still obtain good power.

Round 2
Reviewer 1 Report
The study manuscript appears to have been revised with significant improvements addressing this reviewer’s concerns and comments. There is a need to diversify and improve data visualization and presentation. Further review is left to the discretion of the authors.
Author Response
Thank you for your insightful suggestions. We have polished the expression of the full text and displayed our results more diversely. Considering the significance of the results, we adjusted the color matching of the figures in the results of the subgroup analysis of areas and smoking and replaced figures with tables in the results of the occupation-related subgroup analyses.